# Vitamin C protects retinal ganglion cells via SPP1 in glaucoma and after optic nerve damage

Song Li[1,2], Tatjana C Jakobs[1,2]

**Glaucoma is a common neurodegenerative disorder characterized by retinal ganglion cell death, astrocyte reactivity in the optic nerve, and vision loss. Currently, lowering the intraocular pressure (IOP) is the first-line treatment, but adjuvant neuroprotective approaches would be welcome. Vitamin C possesses neuroprotective activities that are thought to be related to its properties as a co-factor of enzymes and its antioxidant effects. Here, we show that vitamin C promotes a neuroprotective phenotype and increases gene expression related to neurotropic factors, phagocytosis, and mitochondrial ATP production. This effect is dependent on the up-regulation of secreted phosphoprotein 1 (SPP1) in reactive astrocytes via the transcription factor E2F1. SPP1+ astrocytes in turn promote retinal ganglion cell survival in a mouse model of glaucoma. In addition, oral administration of vitamin C lowers the IOP in mice. This study identifies an additional neuroprotective pathway for vitamin C and suggests a potential therapeutic role of vitamin C in neurodegenerative diseases such as glaucoma.**

## Introduction

Glaucoma refers to a group of diseases that lead to the degeneration of retinal ganglion cells (RGCs), their axons in the optic nerve, and the concomitant decline in visual function (Quigley, 2011). Positive family history, elevated intraocular pressure (IOP), and especially age are the main risk factors (Tham et al, 2014). The only modifiable of these is the IOP, and all current glaucoma therapies aim at lowering the IOP by pharmacological or surgical means (Garway-Heath et al, 2015). However, this is not effective in all cases and visual field defects can progress even despite apparently well-controlled IOP. Therefore, the search for alternative neuroprotective therapeutic approaches is ongoing. In recent years, several studies have demonstrated that this is possible, at least in animal models. Possible approaches are inhibiting the neuro-inflammatory response that accompanies glaucomatous changes in the retina and the optic nerve (Bosco et al, 2008; Cueva Vargas

et al, 2016), preventing the entry of monocytes from the bloodstream (Howell et al, 2012; Williams et al, 2019), or interfering with the inflammasome pathway (Krishnan et al, 2016, 2019). Other interventions include the administration of neurotropic factors or antioxidants (Pease et al, 2009; Yang et al, 2016; Pham et al, 2022; Lazaldin et al, 2023), improving mitochondrial function (Ju et al, 2022; Quintero et al, 2022), and preventing the detrimental changes that are common to glaucomatous and age-related degeneration of RGCs (Lu et al, 2020; Xu et al, 2022). In addition to pharmacological treatment, some interventions that fall under the category of lifestyle changes show promise in glaucoma therapy. The benefits of physical exercise for slowing ganglion cell degeneration (and, more generally, neurodegeneration in all parts of the CNS) are well documented (Ong et al, 2018; Chu-Tan et al, 2022; Lopez-Ortiz et al, 2022; Zhang et al, 2022). Dietary intake of vitamins may also have a place in the prevention of neurodegeneration. Vitamin B3 (nicotinamide) has been shown to prevent RGC degeneration in the DBA/2J mouse model of hereditary glaucoma (Williams et al, 2017, 2018), and lower serum levels of nicotinamide are found in human patients with primary open-angle glaucoma (Kouassi Nzoughet et al, 2019). Nicotinamide exerts its effect at least in part by enhancing mitochondrial health and oxidative phosphorylation (Tribble et al, 2021a, 2021b). Other vitamins that may have neuroprotective activity in glaucoma are vitamins A (retinoic acid) and C (ascorbic acid) (Ramdas et al, 2018). This suggests that especially dietary and lifestyle factors that influence mitochondrial function—such as vitamin B3 supplementation or exercise—may be effective adjunct therapies for glaucoma (Williams et al, 2017).

We recently found that the overexpression of the cytokine SPP1 (secreted phosphoprotein 1, osteopontin) in the retina and optic nerve head protects RGCs and rescues visual function in mouse models of aging and glaucoma (Li & Jakobs, 2022). We sought to screen and identify small molecules that increase SPP1 expression as an adjuvant glaucoma treatment. Both vitamins A and C have been described to regulate SPP1 expression in osteoblast and other cell types (Harada et al, 1995; Hadzir et al, 2014; Jeradi & Hammerschmidt, 2016; Nam et al, 2019). In addition, the promoter region of *Spp1* contains retinoic acid response elements. Vitamin C, an essential nutrient for humans, fulfills a number of physiological functions, the prevention of scurvy being the most well-known.

---

[1]Department of Ophthalmology, Harvard Medical School, Boston, MA, USA [2]Schepens Eye Research Institute, Massachusetts Eye and Ear, Boston, MA, USA

Correspondence: song_li@meei.harvard.edu; tatjana_jakobs@meei.harvard.edu

However, recently vitamin C also was found to play an important role in the development of the nervous system (Tveden-Nyborg, 2021; Coker et al, 2022), synaptogenesis (Moretti & Rodrigues, 2021), and oligodendrocyte differentiation (Guo et al, 2018). At least in animal models of neurodegenerative diseases, vitamin C has shown promise as a neuroprotectant (Kangisser et al, 2021).

The evidence for the use of vitamin C as adjuvant therapy in glaucoma is mixed. A number of studies from the 1960s and 1970s reported IOP-lowering effects of intravenous vitamin C, but these effects are most likely due to creating an osmotic gradient between the blood and the intraocular fluids, and not due to the biochemical activities of the vitamin (Linner, 1969; Fishbein & Goodstein, 1972). As the effect only lasts for 12 h, it is of limited practical use. Whether oral vitamin C supplementation has an IOP-lowering effect in humans is controversial (Yuki et al, 2010; Ramdas et al, 2018; Hysi et al, 2019; Han & Fu, 2022). However, independent of IOP, ascorbic acid may be neuroprotective because of its antioxidant properties (Xu et al, 2014) or the activation of ten–eleven translocation enzymes (TETs) (Blaschke et al, 2013).

Here, we report an additional neuroprotective pathway for vitamin C that involves the cytokine SPP1 and the transcription factor E2F1. We show that vitamin C stimulates the astrocytic expression of the SPP1 and induces a neuroprotective phenotype in astrocytes in vitro and in vivo. In addition, vitamin C lowers the IOP and protects RGCs and visual function in a mouse model of glaucoma.

## Results

### Vitamin C up-regulates SPP1 expression in astrocytes

We recently reported that the cytokine-like protein SPP1 is highly protective of RGCs and visual function in several models of injury (elevated IOP, optic nerve crush, and aging) (Li & Jakobs, 2022). In search of small molecules that can up-regulate SPP1 expression, we identified vitamins A (retinoic acid) and C that have been described to regulate SPP1 expression in osteoblast and other cell types (Harada et al, 1995; Hadzir et al, 2014; Jeradi & Hammerschmidt, 2016; Nam et al, 2019). In addition, the promoter region of *Spp1* contains retinoic acid response elements. In cultures of the optic nerve and retinal astrocytes, 1–100 $\mu$M retinoic acid up-regulated SPP1 protein expression in a dose-dependent manner (Figs 1A–C and S1A). *Spp1* mRNA expression was increased by more than twofold in response to 100 $\mu$M retinoic acid (Fig 1D).

We next tested vitamin C and the sodium salt of vitamin C (sodium ascorbate, NaAscorbate). Vitamin C increased SPP1 protein expression in a dose-dependent manner from 31.25 to 250 $\mu$M (Figs 1E–G and S1B). We also found a significant increase in *Spp1* mRNA after stimulation with vitamin C compared to control conditions (Fig 1H). As vitamin C is a weak acid, we also tested sodium ascorbate. Sodium ascorbate enhanced SPP1 protein expression in a dose-dependent manner from 31.25 to 250 $\mu$M (Figs 1I–K and S1C). There was an increased level of SPP1 mRNA in cultured astrocytes treated with 250 $\mu$M sodium ascorbate compared to the vehicle group (Fig 1L). The maximum effect on the increase in SPP1 expression was

similar between vitamin C and sodium ascorbate, but the dose for the maximum effect was lower in sodium ascorbate (62.5 $\mu$M) than that in vitamin C (125 $\mu$M). Sodium ascorbate did not significantly change the pH of the cell culture medium, which was 7.40–7.47 for concentrations between 31.25 and 1,000 $\mu$M, excluding the effects of pH on gene transcription in astrocytes (Fig 1M).

Because *Spp1* gene expression is regulated by transcription factors RUNX1 and E2F1 in astrocytes (Li & Jakobs, 2022), we further sought to test whether these transcription factors are involved in the up-regulation of *Spp1* by vitamin C. Sodium ascorbate up-regulated *Spp1* and *E2f1*, but not *Runx1*, mRNA expression in cultured astrocytes (Fig 1N). The effect on *Spp1* expression was dependent on E2F1, as it was blocked by the E2F1 inhibitor HLM006474 (Fig 1O). Taken together, the results demonstrate that vitamin C up-regulated SPP1 expression via E2F1 in astrocytes.

### Vitamin C induces neuroprotective astrocytes via SPP1

We found previously that SPP1 induces a neuroprotective phenotype in astrocytes, including the inhibition of neurotoxic mediators, promotion of synaptogenic, phagocytic, and neurotropic markers, and an up-regulation of oxidative phosphorylation and ATP production (Li & Jakobs, 2022). To further understand the molecular mechanism of vitamin C activity in astrocytes, we used qRT-PCR to test whether vitamin C could also induce neuroprotective astrocytes by regulating these pathways via SPP1. Wild-type astrocytes in culture responded to sodium ascorbate with a moderate up-regulation of genes related to synaptogenesis, oxidative phosphorylation, neurotropic factors, and phagocytosis-related genes, and a down-regulation of toxic factor Il1$\alpha$, whereas these effects were blocked by *Spp1* deletion in astrocytes (Fig 2A).

A recent publication used sets of marker genes to classify states of reactive astrocytes: a set of pan-reactive genes commonly expressed in reactive astrocytes, a set of "A1" genes that correspond to a neurotoxic astrocyte phenotype, and "A2" genes that characterize predominantly neuroprotective astrocytes (Liddelow et al, 2017). In astrocytes treated with sodium ascorbate, there was a very slight change in A1 neurotoxic markers, but almost all A2 neuroprotective markers were up-regulated (Fig 2B). However, in *Spp1* KO astrocytes, the ascorbate-induced increase in A2 neuroprotective genes was absent, except for the Slc10a6 and Cd109 genes that were still up-regulated in the KO mice. This suggests that in astrocytes, vitamin C induces the up-regulation of neuroprotective markers at least partially via the up-regulation of SPP1 (Fig 2C).

Collectively, vitamin C increases the production of neurotropic factors, and leads to an up-regulation of genes related to phagocytosis and oxidative phosphorylation and induces the gene expression of neuroprotective factors in astrocytes in an SPP1-dependent manner.

### Vitamin C increases SPP1 expression in RGCs in vivo

Because vitamin C up-regulated SPP1 in astrocytes in vitro, we then tested whether vitamin C increases SPP1 expression in the retina in vivo. SPP1 was specifically expressed in alpha ganglion cells in

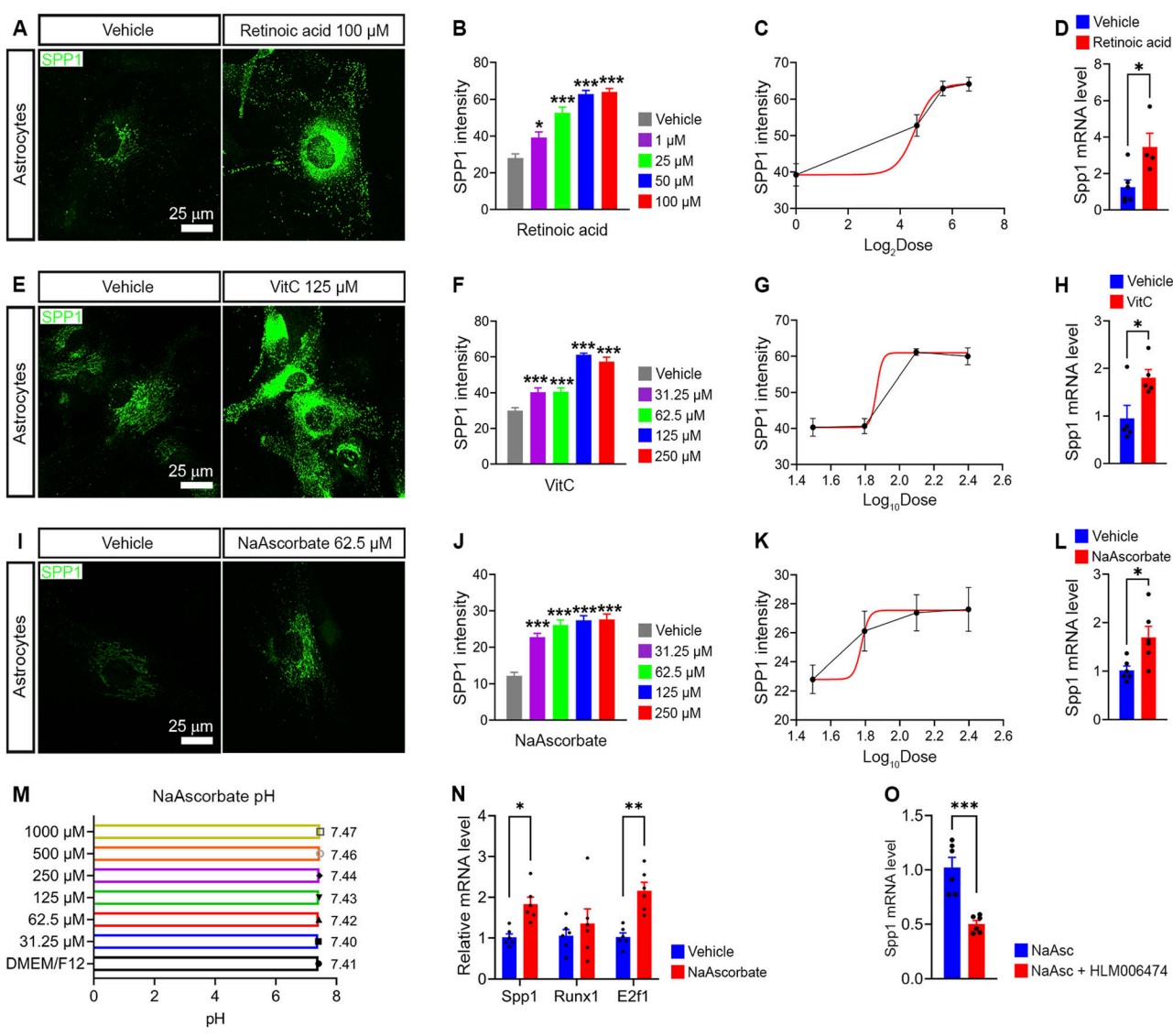

**Figure 1. Vitamin C increases SPP1 expression via E2F1 in astrocytes in vitro.**
**(A)** SPP1 immunocytochemical staining in C57BL/6 astrocytes treated with 100 μM vitamin A (retinoic acid) for 24 h. **(B)** Vitamin A increased SPP1 protein expression in a dose-dependent manner in C57BL/6 astrocytes (n = 14–36 cells from multiple independent wells/group). **(C)** Dose–response curves showed the effect of vitamin A concentration (Log₂Dose) on SPP1 protein expression in C57BL/6 astrocytes (n = 14–36 cells from multiple independent wells/group). **(D)** *Spp1* mRNA was increased by 100 μM vitamin A in C57BL/6 astrocytes (n = 4–6). **(E)** SPP1 immunostaining in C57BL/6 astrocytes treated with 125 μM vitamin C (ascorbic acid) for 24 h. **(F)** Vitamin C up-regulated SPP1 protein expression in a dose-dependent manner in C57BL/6 astrocytes (n = 27–123 cells from multiple independent wells/group). **(G)** Dose–response curves showed the effect of vitamin C concentration (Log₁₀Dose) on SPP1 protein expression in C57BL/6 astrocytes (n = 27–123 cells from multiple independent wells/group). **(H)** *Spp1* mRNA was increased by 125 μM vitamin C in C57BL/6 astrocytes (n = 5). **(I)** SPP1 immunostaining in C57BL/6 astrocytes treated with 62.5 μM NaAscorbate (sodium ascorbate) for 24 h. **(J)** NaAscorbate increased SPP1 protein expression in a dose-dependent manner in C57BL/6 astrocytes (n = 11–31 cells from multiple independent wells/group). **(K)** Dose–response curves showed the effect of NaAscorbate concentration (Log₁₀Dose) on SPP1 protein expression in C57BL/6 astrocytes (n = 11–31 cells from multiple independent wells/group). **(L)** *Spp1* mRNA was up-regulated by 62.5 μM NaAscorbate (n = 6). **(M)** NaAscorbate pH in different concentrations between 31.25 and 1,000 μM showed that the pH was almost neutral from 7.40 to 7.47. **(N)** NaAscorbate increased *Spp1* and *E2f1* mRNA expression, not *Runx1* in cultured C57BL/6 astrocytes (n = 6). **(O)** Increased *Spp1* mRNA expression by NaAscorbate was blocked by E2F1 inhibition with HLM006474 (n = 6). **(B, D, F, H, J, L, O, N)** *P*-values by a one-way ANOVA (B, F, J), an unpaired two-tailed *t* test (D, H, L, O), and a two-way ANOVA (N), *P < 0.05, **P < 0.01, and ***P < 0.001. Data are the mean ± SEM. Source data are available for this figure.

the retina, as SPP1 was co-expressed with the marker SMI32 (neurofilament H) that is most prominently expressed in alpha RGCs (Fig 3A and B). Vitamin C up-regulated SPP1 protein expression in the intrinsically SPP1⁺ alpha ganglion cells in the retina in a dose-dependent manner from 0.1 to 30 μg/μl (Fig 3C, E, and F). We also

detected an obvious increase in SPP1 immunoreactivity in dendrites and axons of ganglion cells at high doses (30 μg/μl) of vitamin C (Fig 3C). Similarly, sodium ascorbate also increased SPP1 expression in RGCs, especially at the 1 μg/μl dose (Fig 3D and G). Taken together, the experiments in vivo demonstrated that vitamin C

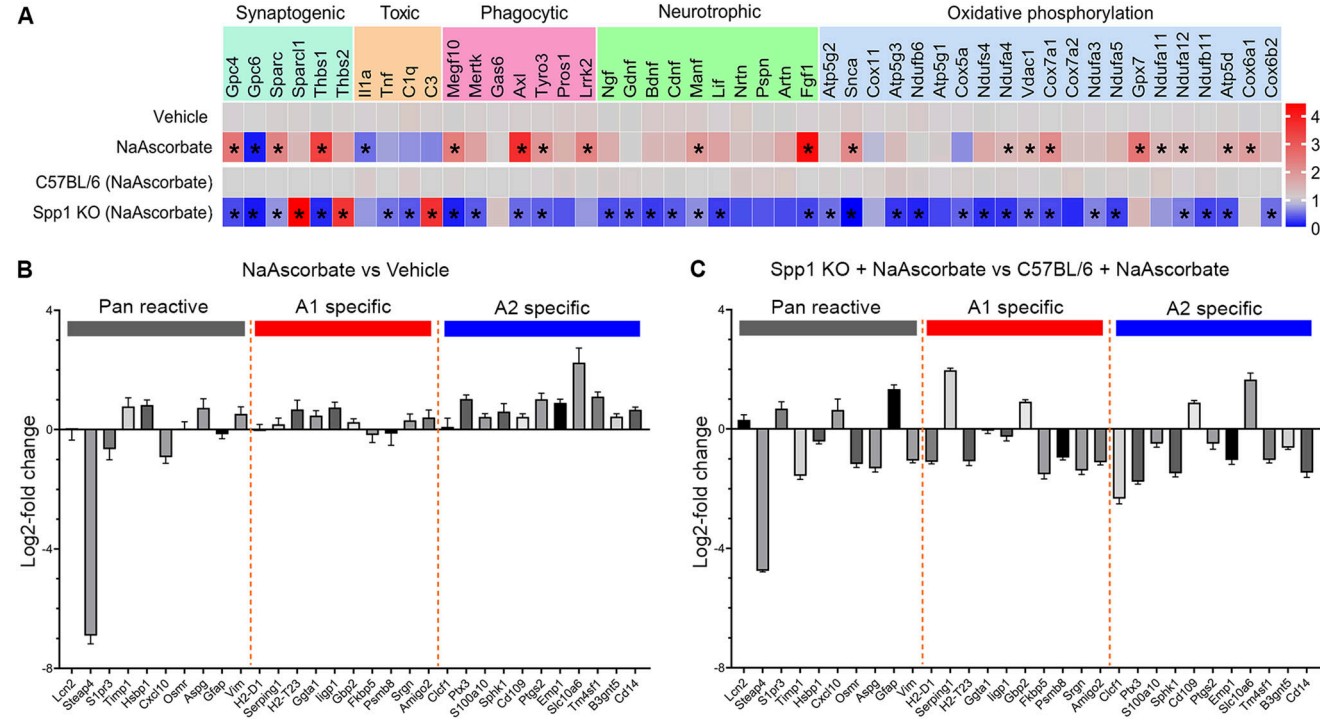

**Figure 2. Vitamin C increases *Spp1* expression to promote astrocytic neuroprotection.**
**(A)** Heat map of expression levels of genes related to synaptogenesis, neurotoxic factors, phagocytosis, neurotropic factors, and oxidative phosphorylation in cultured astrocytes treated with vitamin C (n = 4–6). The gene levels were assessed by qRT-PCR. The pairwise comparisons are between vehicle and NaAscorbate in cultured C57BL/6 astrocytes, and between NaAscorbate-treated C57BL/6 and NaAscorbate-treated *Spp1* KO astrocytes. The asterisks indicate significance with $P < 0.05$ in the pairwise comparisons. **(B)** Effects of 62.5 µM NaAscorbate on gene expression related to astrocyte reactivity in C57BL/6 astrocytes, suggesting NaAscorbate increased "A2" genes. **(C)** Effects of 62.5 µM NaAscorbate on gene expression related to astrocyte reactivity in *Spp1* KO astrocytes, suggesting *Spp1* deficiency prevented the NaAscorbate-induced increase in "A2" genes. **(A)** P-values by a multiple unpaired t test (A), *P < 0.05. Data are the mean ± SEM.
Source data are available for this figure.

and its salt increased SPP1 expression in ganglion cells in the retina.

## Vitamin C promotes astrocytic neuroprotection via the up-regulation of SPP1 in vivo

To test the function of vitamin C in glaucoma, we used the microbead occlusion model in 3-mo-old mice to induce unilateral elevated IOP. Because astrocytic SPP1 was found to be protective for RGCs and visual function (Li & Jakobs, 2022), we then verified this effect in vivo by administering vitamin C (1%) to mice in the drinking water for 2 wk. Vitamin C up-regulated SPP1 protein expression in the intrinsically SPP1+ alpha ganglion cells in the retina and astrocytes in the optic nerve in glaucomatous mice in vivo (Fig 4A and B).

Because vitamin C induced neuroprotective astrocytes in vitro via SPP1, we further investigated the molecular mechanisms of vitamin C in astrocytes in vivo in a mouse glaucoma model. B6.*Gfap*-cre, a strain expressing Cre recombinase under the control of the astrocyte-specific *Gfap* promoter, was crossed with a tdTomato reporter strain (Ai14) to get *Gfap*-tdTomato, in which astrocytes were labeled with a red fluorescent protein. We also used an astrocytic conditional *Spp1* knock-out strain, *Spp1*GFPfl/fl*Gfap*Cre (*Spp1* cKO),

generated by crossing *Spp1*GFPfl/fl with B6.*Gfap*-cre. In addition to conditional deletion of *Spp1*, this strain also expresses tdTomato in astrocytes (Li & Jakobs, 2022). The IOP was raised unilaterally in mice that were treated with vitamin C.

2 wk after induction of high IOP, astrocytes from the optic nerve were isolated. We then used fluorescence-activated cell sorting (FACS) to sort optic nerve head SPP1+ astrocytes from *Gfap*-tdTomato and SPP1− astrocytes from *Spp1*GFPfl/fl*Gfap*Cre mice (Fig 4C and D). As in cell culture, vitamin C led to an up-regulation of *Spp1* and *E2f1* mRNA, but had no effect on *Runx1* expression (Fig 4E). Besides, vitamin C strongly increased gene levels associated with phagocytosis and neurotropic factors, and moderately up-regulated gene expression related to oxidative phosphorylation in optic nerve head astrocytes of glaucomatous mice in vivo. These effects induced by vitamin C were blocked in *Spp1* cKO mice (Fig 4F).

## Vitamin C protects RGCs and rescues vision function in glaucoma

Finally, we asked whether oral supplementation of vitamin C would protect RGCs and visual function in the microbead model of glaucoma. We administered vitamin C (1%) to glaucomatous mice in the drinking water for 2 wk. Age-, strain-, and sex-matched control animals received normal water. 2 wk later, RGC density, pattern ERG,

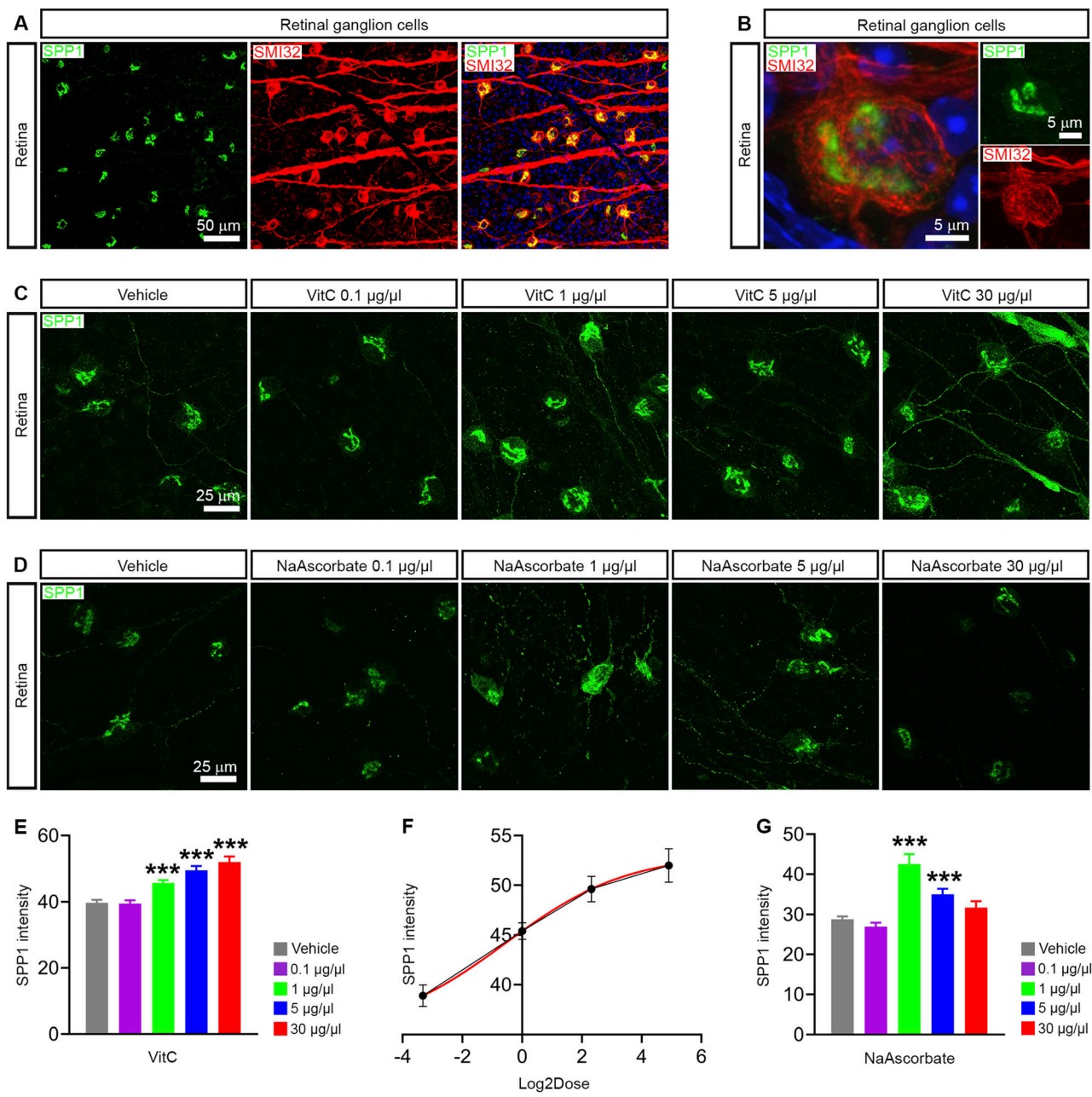

**Figure 3. Vitamin C increases SPP1 expression in the retina in vivo.**
**(A)** Co-immunostaining of SPP1 and alpha retinal ganglion cell marker SMI32 in the C57BL/6 retina. **(B)** High-magnification image of SPP1 immunostaining in the alpha retinal ganglion cell of the C57BL/6 retina. **(C)** SPP1 immunostaining in the C57BL/6 retina treated with 1, 25, 50, and 100 $\mu$M vitamin A (retinoic acid) for 24 h. **(D)** SPP1 immunostaining in the C57BL/6 retina treated with 31.25, 62.5, 125, and 250 $\mu$M vitamin C (ascorbic acid) for 24 h. **(E)** Vitamin C up-regulated SPP1 protein expression in a dose-dependent manner in the C57BL/6 retina ($n$ = 29–103 cells/group). Retinal ganglion cells were from four mice. **(F)** Dose–response curves showed the effect of vitamin C concentration (Log$_2$Dose) on SPP1 protein expression in the C57BL/6 retina ($n$ = 29–103 cells/group). Retinal ganglion cells were from four mice. **(G)** NaAscorbate increased SPP1 protein expression in the C57BL/6 retina ($n$ = 26–48 cells/group). Retinal ganglion cells were from three mice. **(E, G)** $P$-values by a one-way ANOVA (E, G), ***$P$ < 0.001. Data are the mean ± SEM.
Source data are available for this figure.

and visual acuity were measured. Vitamin C supplementation in the drinking water improved RGC survival, pattern ERG, and visual acuity in the glaucoma models of C57BL/6 and *Spp1* KO mice (Figs 5A–D and S2), but vitamin C also significantly reduced the IOP itself (Fig 5E). This made it difficult to determine to what extent the

vitamin C effect was related to its ability to up-regulate SPP1 and to what extent it was related to the effect on IOP.

To further investigate whether vitamin C protects RGC survival via SPP1, we used an optic nerve crush model as an IOP-independent model of optic nerve damage. Vitamin C was strongly protective of

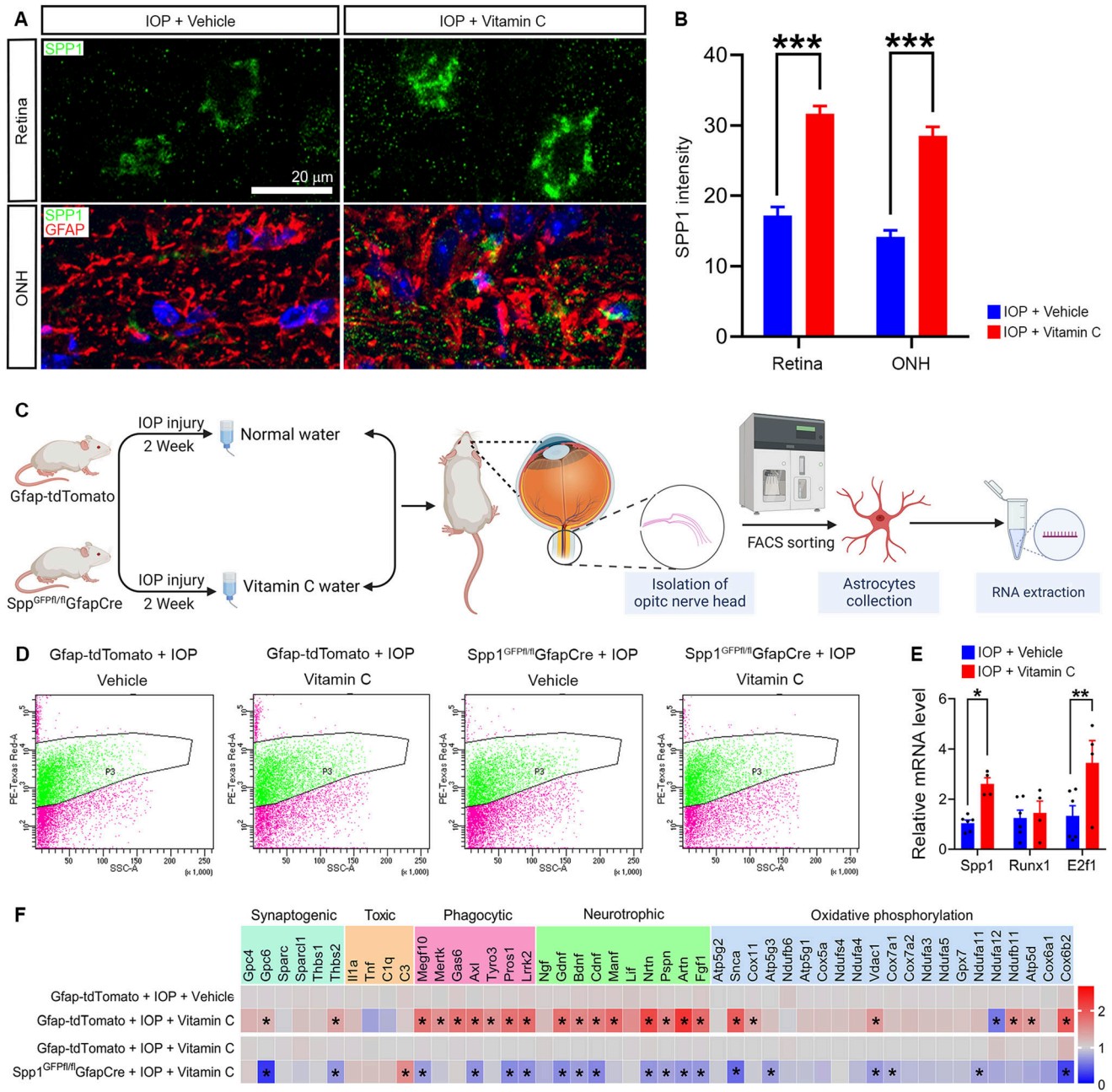

**Figure 4. Vitamin C increases SPP1 expression to promote astrocytic neuroprotection in vivo.**
**(A)** SPP1 immunostaining in the retina and ONH of glaucomatous mice with oral vitamin C treatment. GFAP was used to label astrocytes. **(B)** Vitamin C increased SPP1 protein expression in the retina and ONH of glaucomatous mice in vivo ($n$ = 16–38 cells/group). **(C)** Schematic illustration of tdTomato[+] astrocyte sorting by fluorescence-activated cell sorting from ONH of *Gfap*-tdTomato and *Spp1*[GFPfl/fl]*Gfap*Cre mice treated with oral vitamin C followed by high IOP. **(D)** tdTomato[+] astrocyte sorting from the ONH of Gfap-tdTomato and *Spp1*[GFPfl/fl]*Gfap*Cre mice treated with oral vitamin C followed by high IOP. **(E)** Vitamin C increased *Spp1* and *E2f1* mRNA expression, but not *Runx1* mRNA in sorted ONH astrocytes of glaucomatous mice ($n$ = 4–6). **(F)** Heat map of expression levels of genes related to synaptogenesis, neurotoxic factors, phagocytosis, neurotropic factors, and oxidative phosphorylation in ONH astrocytes sorted from Gfap-tdTomato and Spp1[GFPfl/fl]GfapCre mice treated with oral vitamin C followed by high IOP ($n$ = 4–6). These gene levels were assessed by qRT-PCR. The asterisks indicate significance with $P < 0.05$ in the pairwise comparisons. **(B, E, F)** $P$-values by a two-way ANOVA (B, E) or a multiple unpaired $t$ test (F), *$P < 0.05$, **$P < 0.01$, and ***$P < 0.001$. Data are the mean ± SEM. Source data are available for this figure.

RGC survival in mice with optic nerve crush. This protection in RGC survival induced by vitamin C was absent in *Spp1* KO mice, indicating vitamin C protected RGC survival via SPP1 (Fig 5F and G). We further found vitamin C up-regulated SPP1 protein expression in ganglion cells in the retina and astrocytes in the optic nerve (Fig 5H and I). These results suggest that vitamin C protects RGCs after optic nerve injury partially by up-regulating the expression of the neuroprotective signaling molecule SPP1 (Fig 6).

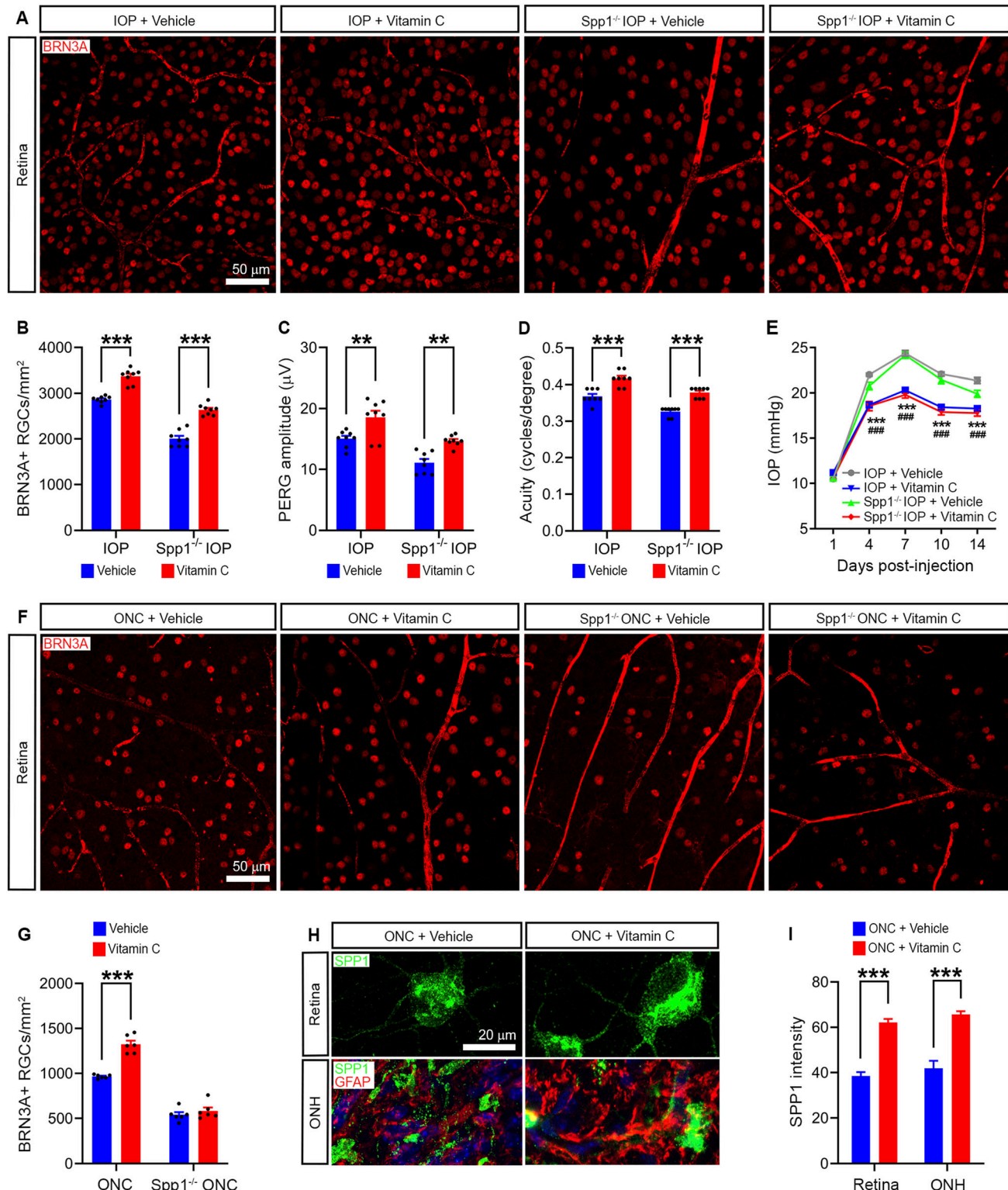

**Figure 5. Vitamin C protects RGCs and vision in a model of glaucoma and ONC.**
**(A)** BRN3A-labeled RGCs in the retina of WT and *Spp1*−/− mice treated with oral vitamin C followed by high IOP. **(B, C, D)** Quantification of the BRN3A+ RGC number (B), PERG amplitude (C), and visual acuity (D) in vitamin C–treated WT and *Spp1*−/− mice with high IOP (*n* = 8). **(E)** Traces of IOP in WT and *Spp1*−/− mice treated with oral vitamin C followed by microbead injection, suggesting vitamin C decreased IOP in glaucomatous mice (*n* = 8). **(F)** BRN3A-labeled RGCs in the retina of WT and *Spp1*−/− mice treated with oral vitamin C followed by ONC. **(G)** Quantification of the BRN3A+ RGC number in the retina of WT and *Spp1*−/− mice treated with oral vitamin C followed by optic nerve crush (*n* = 6). **(H)** SPP1 immunostaining in the retina and ONH of C57BL/6 mice with oral vitamin C treatment followed by optic nerve crush. GFAP was used to label

## Discussion

Vitamin C is abundant in the CNS and fulfills several functions in health and disease (Harrison and May, 2009; Moretti & Rodrigues, 2021). Ascorbate is a co-factor for enzymatic reactions, such as the synthesis of norepinephrine by dopamine-β-hydroxylase (May et al, 2013) and collagen synthesis in blood vessel walls (Sotiriou et al, 2002). The anti-oxidative properties of ascorbate have been linked to its neuroprotective properties in ischemia/reperfusion injury (Huang et al, 2001), in lead poisoning (Han et al, 2007), and possibly in Parkinson's disease (Wagner et al, 1986). Furthermore, ascorbate is directly involved in epigenetic modulation through its interaction with ten–eleven translocation methylcytosine dioxygenases (Blaschke et al, 2013). Here, we show that vitamin C also uses an additional neuroprotective pathway that involves the up-regulation of the cytokine SPP1 in reactive astrocytes.

Astrocytes react to damage to RGCs or their axons by activating a response that—at least in the early stages of disease—aims at protecting the health of surrounding neurons and their processes (Sun et al, 2017). Identifying astrocyte-derived factors that mediate this protective effect may be useful as therapeutics that act via an IOP-independent mechanism and can be added to the pressure-lowering drugs that currently are the first-line treatment for glaucoma. SPP1 was identified in gene expression studies from the optic nerve and other brain regions (Howell et al, 2011; Zamanian et al, 2012; Qu & Jakobs, 2013). Recently, we reported that its overexpression is highly protective of RGC health and visual function in aging and glaucoma (Li & Jakobs, 2022). The clear neuroprotective effect and the apparent safety even in long-term expression studies make SPP1 as an attractive candidate molecule for neuroprotective therapy in glaucoma. However, as a protein, SPP1 would have to be injected intravitreally. Though intravitreal injection is commonly done in anti-VEGF therapy for age-related macular degeneration (Brown & Regillo, 2007), repeated injections carry a risk of inflammation or damage to the retina (Kiss et al, 2018). AAV-mediated overexpression is feasible but has the potential drawback of not being easily reversible once it is initiated. Small molecules that induce SPP1 expression may therefore be useful. The promoter of the Spp1 gene contains predicted binding sites for retinoic acid receptors and T3 (thyroid hormone) receptors. Retinoic acid does increase SPP1 expression—and T3 would be expected to act similarly—but in both cases, the toxic effects of high doses of vitamin A, or T3 overdoses would limit their use. We therefore mainly concentrated on vitamin C.

In cultured wild-type astrocytes, vitamin C up-regulated SPP1 via the transcription factor E2F1. Vitamin C-treated astrocytes also showed an up-regulation of neuroprotective mediators, genes associated with oxidative phosphorylation, and phagocytosis. These effects were dependent on SPP1, as they were not observed in Spp1 KO astrocytes. In the microbead occlusion model of glaucoma, oral supplementation of vitamin C significantly increased visual function

and RGC survival compared with the control group. However, vitamin C also clearly lowered the IOP. As this effect occurred in the wild-type and Spp1 KO mice, it is obviously not mediated by SPP1. It is currently unknown exactly by what mechanisms vitamin C lowers the IOP, but vitamin C affects the trabecular meshwork cells directly, which may improve the aqueous humor outflow facility (Xu et al, 2014).

Though IOP lowering would not be unwelcome in clinical use, in the context of the microbead occlusion model of glaucoma, it acts as a confounding factor as the protection of RGCs and visual function observed in the vitamin C–treated group may be completely due to the IOP-lowering effect. We therefore used optic nerve crush as an IOP-independent model of RGC damage. Optic nerve crush is a much more severe injury than the elevation of IOP and leads to an almost complete loss of RGCs within 2 wk (Berkelaar et al, 1994). Neurotropic factors, including SPP1, can delay, but not completely prevent, RGC loss in this model. Vitamin C was effective in delaying RGC loss in wild-type, but not in Spp1 KO, mice indicating that the neuroprotective effect of vitamin C is at least in part mediated by SPP1.

In the healthy retina and optic nerve, SPP1 is expressed constitutively in ON and OFF alpha ganglion cells, but not in any other type of neurons (Sanes & Masland, 2015). It is only after injury that the astrocytes of the optic nerve (but not those on the retinal surface) and, transiently, microglia express levels of SPP1 that are detectable by immunohistochemistry. Astrocytic SPP1 is a neuroprotective factor (Li & Jakobs, 2022), but the role of SPP1 in alpha ganglion cells is currently unknown. It is tempting to speculate that alpha cells secrete SPP1 in response to injury, either as a signal to glial cells or as a signal to provide direct neuroprotection to surrounding (SPP1-negative) RGCs. Thus, in addition to the direct activity that vitamin C has on RGCs as an antioxidant and a co-factor to enzymes such as TET dioxygenases, a part of the neuroprotection in the ganglion cell layer may be mediated by SPP1.

Most studies of vitamin C in the context of glaucoma have concentrated on the IOP. Interestingly, in human populations, there seems to be no clear-cut effect of vitamin C supplementation on IOP. L-gulono-γ-lactone oxidase catalyzes the last step in the biosynthesis of ascorbic acid. The GULO gene is a pseudogene in humans, other primates, and some other species, which are therefore dependent on a dietary supply of vitamin C. In contrast, mice possess a functional Gulo gene and can synthesize their own vitamin C. In spite of this, supplementing dietary vitamin C is beneficial for mice as demonstrated not only here, but also in other publications that studied parameters such as longevity and neuroprotection. This suggests that mice on standard laboratory chow are probably not optimally supplied with vitamin C. In contrast, vitamin C intake in humans varies with dietary habits. In human subjects who are already optimally supplied, additional intake of vitamin C has no benefit as excess vitamin C is simply excreted via the urine. However, studies that measured serum levels of ascorbate or its metabolites did find negative correlations between serum levels and IOP,

---

astrocytes. **(I)** Vitamin C increased SPP1 protein expression in the retina and ONH of C57BL/6 mice after ONC in vivo ($n$ = 22–27 cells/group). **(B, C, D, E, G, I)** $P$-values by a two-way ANOVA (B, C, D, G, I), *$P$ < 0.05, **$P$ < 0.01, and ***$P$ < 0.001; and a two-way ANOVA (E), ***$P$ < 0.001 (IOP + vehicle versus IOP + vitamin C) and ###$P$ < 0.001 ($Spp1^{-/-}$ IOP + vehicle versus $Spp1^{-/-}$ IOP + vitamin C). Data are the mean ± SEM.
Source data are available for this figure.

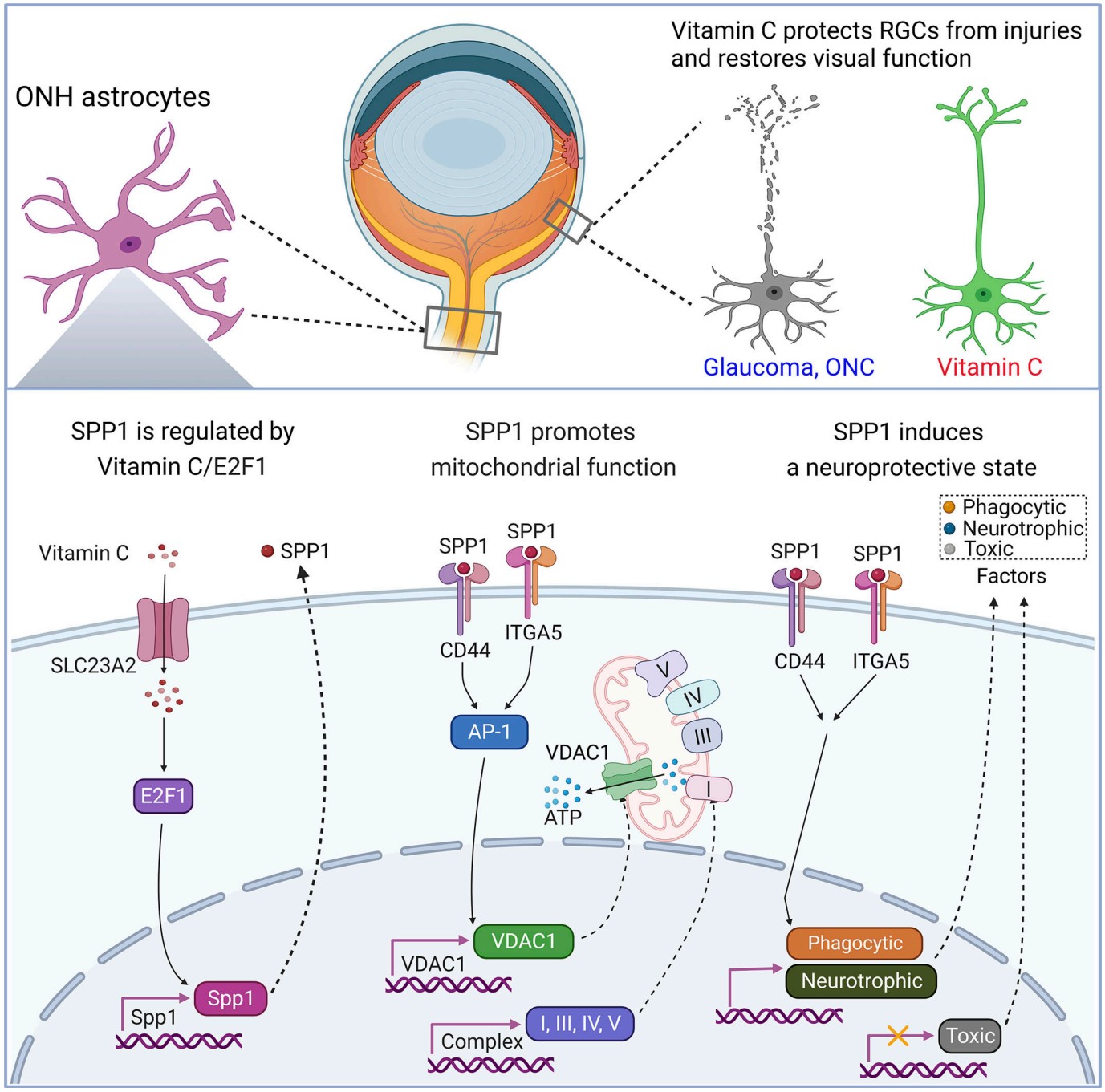

**Figure 6.  Proposed model of vitamin C promotion on RGC survival and visual function.**
Schematic illustration showing vitamin C up-regulates SPP1 expression in reactive astrocytes via the transcription factor E2F1. Vitamin C increases oxidative phosphorylation and VDAC1 expression through SPP1. Vitamin C promotes phagocytosis and the secretion of neurotropic factors in astrocytes but inhibits the production of neurotoxic and inflammatory mediators through SPP1. Loss of SPP1 increases the vulnerability of retinal ganglion cells to injury. Oral administration of vitamin C protects retinal ganglion cells, lowers the IOP, and preserves visual function in glaucoma and traumatic optic injury.

suggesting that there is a subpopulation of glaucoma patients who might benefit (Yuki et al, 2010; Hysi et al, 2019).

The recommended daily intake of vitamin C is 90 mg for adult men and 75 mg for adult women (Institute of Medicine (US) Panel on Dietary Antioxidants & Related Compounds, 2000). There is no proven benefit in exceeding these amounts. Though vitamin C is relatively non-toxic, very high (>2 g per day) doses can lead to gastrointestinal disturbances and are not recommended. Vitamin C is therefore certainly not suitable as a stand-alone treatment for glaucoma. However, our data suggest that suboptimal levels of ascorbate would

exacerbate RGC loss, and taking care to maintain a sufficient intake of vitamin C would be beneficial for glaucoma patients.

# Materials and Methods

### Animals

C57BL/6 (000664) and *Spp1* KO (B6.129S6(Cg)-*Spp1*tm1Blh/J, 004936) mice were purchased from the Jackson Laboratory. *Spp1* KO mice were deficient for SPP1 expression in all tissues. The B6.*Spp1*fl-EGFP-stop-tdTomato strain was described in detail in a previous study (Li & Jakobs, 2022) and was used for the conditional deletion of *Spp1*. B6.Cg-Tg(*Gfap*-cre)77.6Mvs/2J (024098) mice, which were used for creating the astrocyte-specific deletion, and B6.Cg-Gt(ROSA)26Sortm14(CAG-tdTomato)Hze/J (007914) mice, which were used as a reporter strain Ai14 to create *Spp1*+ astrocytes expressing a red fluorescent protein. These mice were purchased from the Jackson Laboratory. Male and female mice were used in equal numbers, and all control mice were strictly age- and sex-matched. Mice were housed under a 12-h dark/light cycle at an ambient temperature of 21–22°C and 40–50% humidity, and received water and standard food ad libitum. Mice were anesthetized by i.p. injection of ketamine and xylazine (100 and 10 mg/kg, respectively), supplemented by topical application of 0.5% proparacaine to the ocular surface. After surgery, mice received an s.c. injection of 0.1 mg/kg buprenorphine. At the end of the experimental period, mice were euthanized by $CO_2$ inhalation according to the Guide for the Care and Use of Laboratory Animals of the AAALAC. All animal care and handling procedures were done in accordance with the guidelines of the Association for Research in Vision and Ophthalmology and approved by the Institutional Animal Care and Use Committee at Schepens Eye Research Institute.

### Primary astrocyte cultures

Retinas and optic nerves from either C57BL/6 wild-type or *Spp1* KO neonatal mice aged 1–3 d were enzymatically digested by 0.25% trypsin and 0.01% DNase at 37°C. The tissue was then triturated by gentle mechanical dissociation with a fire-polished glass Pasteur pipette at room temperature. The suspension was filtered through a 52-$\mu$m nylon mesh filter to generate a single-cell suspension.

Cells were plated on poly-lysine–coated coverslips or tissue culture plates in DMEM/F12 with 10% FBS. Cultures were incubated at 37°C, 95% humidity, and 5% $CO_2$ in a medium. The medium was changed by replacing the fresh medium twice a week. After 7–10 d in culture, the plates were shaken at 200 rpm for 18 h on an orbital shaker, and the supernatant was discarded. This removed most of the oligodendrocytes and microglia. The remaining cells were digested using trypsin and passaged. After two rounds of dissociation and reseeding, more than 95% of the remaining cells were positive for the astrocyte marker GFAP.

### FACS

Astrocytes were purified from the optic nerve head of *Gfap*-tdTomato and *Spp1*GFPfl/fl*Gfap*Cre mice. Briefly, the optic nerve heads were treated with papain (0.6 mg/ml; LS003126; Worthington) and L-cysteine (0.012 mg/ml; C7352; Sigma-Aldrich) for 15 min at 37°C in $Ca^{2+}$- and $Ma^{2+}$-free HBSS solution (14185-052; Gibco). After the incubation, tissues were centrifuged at 953$g$ for 5 min and the papain/HBSS solution was removed. Tissues were resuspended in horse serum (10%, 26050-088; Gibco) and DNase I (60 U/ml; D-5025; Sigma-Aldrich) in HBSS and were triturated with a heat-polished Pasteur pipet (TW150-4; World Precision Instruments), and the tissue was completely dissociated. Dissociated cells were centrifuged, resuspended in HBSS, and passed through a 35-$\mu$m cell strainer. Astrocytes were identified by tdTomato fluorescence and sorted directly into a collection medium using a BD FACSAria III instrument (BD Biosciences). Astrocytes sorted by FACS were centrifuged at 1,000$g$ for 10 min and were used for RNA extraction.

### Electroretinography

Mice were anesthetized, and the pupils were dilated with one drop of 1% tropicamide. Pattern electroretinograms (PERG) were recorded from light-adapted mice on a Celeris small animal testing system (Diagnosys LLC) using a high-contrast horizontal grating (0.05 cycles/degree, reversing in the spatial phase at 1 Hz, 50 cd/$m^2$ mean luminance). Artificial tears (GenTeal) were used to prevent drying and to increase the contact with the recording electrode. An electrode on the contralateral eye served as a reference. A total of 300 complete contrast reversals of pattern ERG were repeated twice in each eye. PERG amplitudes were defined as the difference between the P1 peak and N2. Conventional, light-adapted ERG was recorded directly afterward to ensure that reductions in PERG amplitudes were not due to unrelated ocular disease.

### Optomotor response

The visual acuity of mice was measured based on the optomotor reflex using OptoDrum (Striatech GmbH). In the OptoDrum, wake mice were placed on an elevated platform, surrounded by four computer monitors. A camera observed the behavior of the animal from above. The optomotor reflex was triggered with a rotating black-and-white stripe pattern. Tracking behavior was automatically detected and analyzed by OptoDrum software. Stimulus patterns were continuously and automatically adjusted during the experiment until the visual threshold was reached, and the optomotor reflex was not triggered anymore. The highest spatial frequency that still elicited the reflex was recorded as the visual acuity.

### Retro-orbital optic nerve crush

Mice were anesthetized, and the intraorbital optic nerve was exposed by dissection of the conjunctiva (Sun et al, 2010). The nerve was crushed for 10 s ~1 mm distal to the lamina cribrosa using self-closing jeweler's forceps. 1 wk after surgery, retinas were collected for immunofluorescence to determine the viability of RGCs. Optic nerve heads were used to sort astrocytes by FACS for further qRT-PCR analyses.

### Microbead injection and intraocular pressure measurements

The microbead occlusion model was used to achieve intraocular pressure (IOP) elevation according to published protocols (Sappington et al, 2010; Chen et al, 2011). Mice were anesthetized, and a small hole was made in the cornea. Polystyrene microbeads (15 $\mu$m diameter; Invitrogen) were injected into the anterior chamber of the right eye through the cornea. Control groups received an injection of sterile saline solution. IOPs were measured every 3 d, at the same time of day, using a rebound tonometer (TonoLab; Icare). The reported IOP for each day consists of an average reading of five measurements from each eye. Mice showing no IOP elevation were omitted from the study.

### Tissue preparation

After $CO_2$ euthanasia, the skull was opened, the brain was removed, and eyes and optic nerves were dissected from the surrounding tissue (Jakobs et al, 2005) and immediately fixed in 4% paraformaldehyde overnight. Eyes were hemisected along the ora serrata, and the retinas were removed from the posterior eyecup and whole-mounted on nitrocellulose filters. Optic nerves were left intact and processed for sectioning.

### Immunohistochemistry and immunocytochemistry

After euthanasia, whole eyes were dissected and placed in ice-cold 4% paraformaldehyde. Retinas were mounted RGC side-up on nitrocellulose membranes, blocked with 5% goat serum, and stained for 3 d with primary antibodies (Table S1). Optic nerves were cryoprotected in 30% sucrose, embedded in the OCT compound, and sectioned at 12 $\mu$m in a Leica cryostat. Sections were incubated with primary antibodies overnight at 4°C. Cultured astrocytes plated on poly-lysine–coated coverslips were fixed with 4% formaldehyde for 10 min. After blocking with 5% goat serum, astrocytes were incubated with anti-SPP1 for 1 h. Primary antibodies were visualized with appropriate secondary antibodies conjugated with Alexa fluorophores (Jackson ImmunoResearch and Molecular Probes). DAPI was used to counterstain nuclei. Samples were then mounted in Vectashield (Vector Laboratories). Immunofluorescent images were collected using a Leica SP8 fluorescence microscope (Leica). SPP1 protein expression in the retina in vivo was quantified by the fluorescence intensities of SPP1 immunostaining in SPP1-positive RGCs. SPP1-negative RGCs were not quantified. SPP1-positive RGCs in the acquired images from parts of the retina were used for quantification, and fluorescence intensities of SPP1 immunostaining were analyzed in the plotted region per cell with ImageJ software. The mean fluorescence of SPP1 immunostaining in all quantified RGCs was recorded and used for comparison.

### Confocal microscopy

Confocal image stacks were taken on a Leica SP8 confocal microscope (Leica). For RGC counting, images of the whole-mounted retina were obtained as z-stacks through the RGC layer at a 0.5 $\mu$m step size, and a z-projection was generated.

### cDNA synthesis and quantitative PCR

Because of the small amount of tissue from the optic nerve head, three optic nerve heads were pooled for one sample to sort astrocytes by FACS. One well of cultured astrocytes growing in six-well plates was one sample. Total RNA was isolated from FACS-sorted astrocytes or from cultured astrocytes using RNeasy Plus Micro Kit (QIAGEN). We performed RNA integrity validation and quantification using the Agilent RNA Pico chip analysis in Agilent 2100 Bioanalyzer (Agilent Technologies). Only the RNA samples with an RNA integrity number higher than 7 were used for cDNA synthesis. 10 ng of total RNA purified from astrocytes was reverse-transcribed using the Ovation qRT-PCR system (NuGEN), and the cDNAs were diluted 1:50 as templates to measure transcript levels of candidate genes by quantitative PCR. GAPDH was used as a reference gene as the expression level in the optic nerve head is stable after optic nerve crush (Qu & Jakobs, 2013). Primer sequences are given in Table S1. Gene levels related to synaptogenesis, neurotoxic factors, phagocytosis, neurotropic factors, and oxidative phosphorylation were tested by qRT-PCR. At least five biological replicates were used for each condition, and all samples were run in triplicate with a non-template control on a StepOnePlus qRT-PCR thermocycler (96-Well; Applied Biosystems) and a Roche LightCycler 480 II (384-Well; Roche Diagnostics Corporation).

### Oral administration of vitamin C

Mice were given drinking water supplemented with 1% vitamin C (ascorbic acid), as recommended (Massie et al, 1984). The vitamin C–containing water was prepared every day. Vitamin C administration started on the same day when microbeads were injected. Mice were maintained at the stated dose for 2 wk before euthanasia.

### Statistical analysis

Statistical significance was performed with GraphPad Prism 8. The data are provided as the mean ± SEM. Data were analyzed using an unpaired two-tailed $t$ test for comparisons of two groups. For comparison of multiple groups, normally distributed data were assessed using a one-way ANOVA with Tukey's post-test. For bivariate comparisons, a two-way ANOVA with Bonferroni's post-test was used. Differences were considered statistically significant at $P <$ 0.05. Figures were prepared using Adobe Photoshop CS6. The proposed model (Fig 6) was prepared using BioRender.

# Data Availability

Data are available on reasonable request. Pattern ERG traces are available under Jakobs (2023). All data are available from the corresponding author on request.

# Supplementary Information

# Acknowledgements

This work was supported by the NIH grant R01 EY19703 and the NIH Core Grant for Vision Research P30EY003790, and a Shaffer grant of the Glaucoma Research Foundation.

## Author Contributions

S Li: conceptualization, data curation, formal analysis, investigation, visualization, methodology, and writing—review and editing.
TC Jakobs: conceptualization, data curation, formal analysis, supervision, funding acquisition, investigation, methodology, project administration, and writing—original draft, review, and editing.

## Conflict of Interest Statement

The authors declare that they have no conflict of interest.

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
