## [Reviewer comments · Life Science Alliance]

Life Science Alliance

Vitamin C protects retinal ganglion cells via SPP1 in glaucoma and after optic nerve damage

Song Li and Tatjana Jakobs

DOI: <https://doi.org/10.26508/lsa.202301976>

Corresponding author(s): Tatjana Jakobs, Massachusetts Eye and Ear Infirmary and Song Li, Massachusetts Eye & Ear Infirmary/Schepens Eye Institute

Review Timeline:

Submission Date:	2023-02-07
Editorial Decision:	2023-03-14
Revision Received:	2023-03-31
Editorial Decision:	2023-04-21
Revision Received:	2023-04-29
Accepted:	2023-05-01

Transaction Report:

March 14, 2023

Re: Life Science Alliance manuscript #LSA-2023-01976-T

Dr. Tatjana C Jakobs
Massachusetts Eye & Ear Infirmary/Schepens Eye Institute
Ophthalmology
20 Stanford Str
Boston, MA 2114

Dear Dr. Jakobs,

Thank you for submitting your manuscript entitled "Vitamin C protects retinal ganglion cells via SPP1 in glaucoma and after optic nerve damage" to Life Science Alliance. The manuscript was assessed by expert reviewers, whose comments are appended to this letter. We invite you to submit a revised manuscript addressing the Reviewer comments.

Thank you for this interesting contribution to Life Science Alliance. We are looking forward to receiving your revised manuscript.

Sincerely,

B. MANUSCRIPT ORGANIZATION AND FORMATTING:

Reviewer #1 (Comments to the Authors (Required)):

Summary:

The authors present interesting studies to follow-up recent work describing neuroprotective properties of the secreted phosphoprotein SPP1 from reactive astrocytes in the optic nerve head. Here they assess the use of vitamin C (and derivative) to induce the expression of SPP1, which they argue occurs via the transcription factor E2F1. After some initial experiments to map this pathway in cultured astrocytes, the hypothesis is tested in vivo using two mouse models of ganglion cell injury. The authors conclude that vitamin C induces neuroprotective astrocyte phenotypes, but that these data are hard to interpret in vivo due to independent intraocular (IOP) lowering effects, which would have a similar protective effect. The topic is of interest, as the interplay between astrocytes and neuronal function is very topical, and new neuroprotective pathways are highly sought. In general, the experiments are well presented, and the analyses are rigorous and sound. However, there are details missing throughout, and the overarching hypothesis is not thoroughly tested through these studies. This is due to the pleiotropic nature of vitamin C actions, which are partially addressed by in vitro and conditional knockout studies. The authors note the impact of vitamin C on IOP as a confounding factor, but I feel a more important consideration is potential direct neuroprotective effects, bypassing the astrocytic pathway. This interpretation is supported by the author's own data showing induced neuronal expression of SPP1, but is not given much consideration in the present manuscript. Additional specific points are described below:

Points:

1. What is the rationale for focusing on NaAscorbate treatment in vitro vs Vitamin C? This should be spelled out.
2. Figure 1. What is the "n" for C,G,K panels? Also, the figure lists n=number of cells for the analyses, but please clarify if this was from multiple independent cultures/wells? Also, this is a minor point, but the presentation of panel M exaggerates the small pH differences at first glance, I would suggest a revised presentation that shows the full scale (which will demonstrate the opposite impression)
3. Figure 2. Where do these data come from? There is no description in the methods and it seems unlikely that 48 genes were assessed by standard qPCR (if they were, then the primer sets should be attached as a supplemental file). What do the asterisks indicate in panel A? If this indicates significance, then it should be spelled out.
4. Figure 2. The text describing these data is a little too conclusive regarding A2 vs A1 astrocytic genes, as all seem impacted by both NaAscorbate treatment, and especially Spp1 KO. I suggest tempering this language to better describe the results.
5. Figure 3. Unless I'm misunderstanding, it seems SPP1 is only staining in alpha ganglion cells in the retina in situ, and not astrocytes. What about the optic nerve? Doesn't this suggest that there might be direct neuroprotective effects for this treatment? This should be addressed. Also, how many animals do these data represent? (only #'s of analyzed cells are listed).
6. Figure 4. The colocalization of SPP1 staining in astrocytes of the ONH in panel A is not very convincing, although the sorting data do add alternative evidence. Once again, how were the 48-gene panel data generated?
7. Figure 5. In the graphs B-E, it seems that the protective effect of vitamin C are independent of SPP1. The authors should comment on this observation and compare it to the optic nerve crush model, which does seem to be SPP1 dependent.
8. In the discussion (note the missing heading), the authors focus on the potential for SPP1 to be induced in A2 reactive astrocytes. They also describe the potentially confounding effects of IOP lowering in the microbead model. However, it seems to me another clear possibility is that vitamin C induces direct neuroprotective signaling through RGCs themselves. Their own data in Figure 3 showing induced RGC expression of SPP1 supports this idea. There is also a fairly extensive literature linking vitamin C to a variety of neuroprotective effects. This point should be addressed - Otherwise, the conclusions about astrocytic functions should perhaps be tempered accordingly.

Reviewer #2 (Comments to the Authors (Required)):

This manuscript describes the findings that vitamin C stimulates the astrocytic expression of the SPP1 (secreted phosphoprotein 1, osteopontin), thereby inducing a neuroprotective phenotype in astrocytes in vitro and in vivo. The authors used mouse models (wild type C57BL6) and Spp1 knock-out (B6.129S(Cg)-Spp1tm1Blh/J, 004936) as well as primary astrocyte cultures in their study. The manuscript is well-written and thoughtful, and the research is methodical, thorough, and sound. With minor changes, publication of this manuscript would be valuable to the vision research community. Minor changes include:

- include all data points in dot format
- include representative traces of ERG data
- when introducing vitamin C, include (ascorbic acid) as vitamin A (retinoic acid) (page 5,7)
- expand upon discussion on the results varied between NaAscorbate and VitC

Reviewer #3 (Comments to the Authors (Required)):

Referee Cross-Commenting:

Reviewers 1 and 2 raise some very valid points. I want to highlight the following points from reviewer # 1.

The Ns in some of the plots are missing and should be stated, e.g. in Fig 1 C, G, K. The number of animals/eyes should be stated for plots from in vivo experiments where only the number of cells are stated (e.g. in Fig 3).

The supplemental data was missing from the merged manuscript pdf due to no fault of the authors I think. It does contain the primer sequences for the 48 genes that Reviewer #1 was looking for.

1. A short summary of the paper, including description of the advance offered to the field.

In this manuscript, Li and Jakobs investigated the role of oral supplementation of vitamin C on preventing neurodegeneration and preserving visual function in mouse models of glaucoma. Using well designed, careful experiments they honed in on a particular mechanism which could be behind the beneficial effects of vitamin C supplementation. Building on their previous work on the cytokine Spp1 (Li and Jakobs 2022), in this manuscript they show that vitamin C exerts a beneficial effect by inducing a neuroprotective class of reactive astrocytes both in well designed in vitro experiments as well as in in vivo models of glaucoma in mice. I think the paper is carefully written and the experiments in general are well designed and executed. The authors took care to utilize age, sex and strain matched controls in their in vivo experiments. I also really commend the authors for making the figures with great care. The data was very well presented. The discussion section of the paper is particularly well written and addresses important caveats including some that I thought of while reading the paper. The discussion on Vitamin C supplementation in humans vs. mice is really useful. I do think that the authors somewhat overstate the results for visual function in the abstract but overall its solid work.

I think this paper certainly deserves to be published in Life Science Alliance, with the revisions to the text I requested in 3.

2. For each main point of the paper, please indicate if the data are strongly supportive. If not, explicitly state the additional experiments essential to support the claims made and the timeframe that these would require.

The paper can be broadly divided into two subsections.

The authors start out with crucial in vitro experiments that lay the foundation for their subsequent in vivo experiments in mice.

In the in vitro section, building on their prior work, the authors demonstrate that vitamin C as well as ascorbate can induce Spp1 expression in cultured astrocytes via the transcription factor E2f1 in a dose dependent manner (figure 1). The Spp1 induction in astrocytes is central to the authors' hypothesis on how Vitamin C exerts its protective effects, them having recently published a paper on this topic (Li and Jakobs 2022). The representative images in Figure 1 are convincing, the data is presented in a way that's easy to follow. In Figure 2, the authors utilize the findings from a landmark paper by Ben Barres' group in 2017 (Liddel et al. 2017) showing that neuroinflammation and ischemia can induce two distinct category of reactive astrocytes, a neurotoxic class termed A1 and a neuroprotective class A2 and where class is characterized by their characteristic up/down regulation of key gene markers among other differences. In this manuscript, Li and Jakobs demonstrate that in vitro vitamin C supplementation induces markers in astrocytes that classify them closer to the A2 group of neuroprotective astrocytes in the Barres Lab paper.

In the next part of the paper, the authors investigate the protective role of Vitamin C in *in vivo* models of glaucoma in mice. The authors show that Vitamin C supplementation induces *Spp1* expression in the retina, oddly enough in a subclass of retinal ganglion cells (RGCs): alpha. The fluorescence images look reasonably convincing. In Figure 4, the authors show that vitamin C oral supplementation induces *Spp1* in RGCs and optic nerve head (ONH) astrocytes in an elevated IOP model in mice. Then, using an elegant technique the authors isolate the *Spp1*⁺ astrocytes from WT and *Spp1* KO mice subjected to elevated intraocular pressure (IOP), and show that astrocytes in vitamin C treated mice but not vehicle treated mice exhibit elevated *Spp1* as well as the transcription factor *E2f1* in keeping with their results from *in vitro* experiments. Further, the astrocytes from Vitamin C treated IOP mice exhibit a gene expression profile that's putatively conducive to neuroprotection. This is also blocked in *Spp1* KO mice showing that this effect of vitamin C *in vivo* in the glaucomatous mice is *Spp1* dependent. This figure is really well presented and the experiments are very carefully designed and clever. In the next figure the authors seek to demonstrate the beneficial effects of Vitamin C oral supplementation on the elevated IOP model in WT and *SPP1* KO mice. Although Vitamin C appears to be neuroprotective (figure 5B) and appears to preserve visual function (figure 5C, 5D) in the IOP animal model in WT mice, interestingly the *SPP1* KO mice subjected to IOP and treated with vitamin C also appear to benefit from a neuroprotective effect (figure 5B) and are able to preserve visual function (5C, 5D). The authors discovered that Vitamin C also causes a downregulation of IOP (figure 5E) and this thus presents a confounding factor. They navigate this issue by utilizing another model of RGC loss that is IOP independent, optic nerve crush (ONC). In figure 5G, even though the WT mice treated with vitamin C have higher RGC counts than vehicle, the *Spp1* KO mice do not exhibit any difference between Vitamin C and vehicle treated animals. The authors state that vitamin C also reduced the IOP which could be responsible for the beneficial effects rather than *Spp1* induction. It might be helpful to highlight here or in the discussion if this could potentially explain why the *Spp1* KO mice in Figure 5B, 5C, 5D show a similar effect of Vitamin C treatment as the WT mice.

3. Lastly, indicate any additional issues you feel should be addressed (text changes, data presentation, statistics etc.).

Important:

- a. The sentence in the abstract (lines 5 through 7) "Here we show that vitamin C promotes a neuroprotective phenotype in reactive astrocytes by increasing the release of neurotrophic factors, phagocytosis, and mitochondrial ATP production." overstates what the authors have done in this manuscript. I appreciate that the authors did functional assays for mitochondrial ATP production and phagocytosis in their previous paper on *Spp1* (Li and Jakobs 2022). But this sentence misleadingly gives the impression that similar work was performed in the current manuscript when the only experiments presented in this work are gene expression assays for markers of mitochondrial function and phagocytosis. I do not think that the authors need to do the functional experiments. However, the sentence in the abstract should be revised to accurately reflect the work that was done in this specific paper.
- b. I commend the authors for discussing some of the caveats of the effect of Vitamin C on IOP. However, I do think that based on the constraints of the experiments in figure 5, the authors cannot conclude that *Spp1*⁺ astrocytes are causing preservation of visual function. This is especially because the *Spp1* KO mice in Figure 5C and 5D also exhibited improved visual function when treated with Vitamin C. If the preservation of visual function by Vitamin C is *Spp1*⁺ astrocyte dependent then the data from *Spp1* KO mice in figures 5C and 5D contradict this claim. The only beneficial effect they can conclude for certain is the neuroprotective effect based on the *Brn3a*⁺ RGC count in figure 5G where the KO mice do not show any effect due to Vitamin C supplementation. As visual function is not assessed in the ONC model after vitamin C supplementation, the claim about visual function should be proportionate to the results presented. Thus, the appropriate parts of the manuscript should be revised to reflect this.
i.e., in the abstract (lines 9 through 10): "*SPP1*⁺ astrocytes in turn promote RGC survival and improve visual function in a mouse model of glaucoma.". They can revise it to say that "Vitamin C promotes RGC survival and visual function" or "*SPP1*⁺ astrocytes in turn promote RGC survival."
- c. It would be helpful if the authors can include in the methods/figure legends more details about how the fluorescence intensities were quantified and plotted, especially for the *in vivo* experiments like in figure 3, whether the *Spp1* intensities per cell for all *SMI32*⁺ cells in the whole retina were quantified or some parts (which parts) etc. Assuming that the intensity plotted is per cell and the error bars are over cells, it would be important to know these details in order to better assess how robust the effect of Vitamin C or ascorbate supplementation is on a per animal basis. Moreover, it would be great if they could clarify how they handled *Spp1* negative cells in this quantification. Were *Spp1* negative cells included in the calculation? So lets say, hypothetically speaking a cell was negative for *Spp1* in the vitamin C treatment group (even though it is *SMI32*⁺) would that have been included in the means calculation for the group? In figure 3E, the legend says n = 29-103 cells/group. It would be very helpful to state how the authors chose how many, which cells to count, in the whole mount retina for plots like this. Obviously, based on the numbers it does not look like the whole retina was quantified(?).

Minor comments:

a. Do both OFF and ON alpha RGCs express Spp1? What, if any is the role of Spp1 in RGCs?

b. An RNAseq approach would have been better for figure 4f, especially since it may have helped generate more insight about the Vitamin C induced, neuroprotection mechanism downstream of Spp1 and potentially also generated putative small molecule targets. RNAseq on retinal cells after Vitamin C treatment in Spp1 KO vs WT animals might also be useful for future studies.

March 30, 2023

Re: Life Science Alliance manuscript #LSA-2023-01976-T

We thank the reviewers for the detailed and helpful comments. Our answers are underlined.

Reviewer 1:

Summary:

The authors present interesting studies to follow-up recent work describing neuroprotective properties of the secreted phosphoprotein SPP1 from reactive astrocytes in the optic nerve head. Here they assess the use of vitamin C (and derivative) to induce the expression of SPP1, which they argue occurs via the transcription factor E2F1. After some initial experiments to map this pathway in cultured astrocytes, the hypothesis is tested in vivo using two mouse models of ganglion cell injury. The authors conclude that vitamin C induces neuroprotective astrocyte phenotypes, but that these data are hard to interpret in vivo due to independent intraocular (IOP) lowering effects, which would have a similar protective effect. The topic is of interest, as the interplay between astrocytes and neuronal function is very topical, and new neuroprotective pathways are highly sought. In general, the experiments are well presented, and the analyses are rigorous and sound. However, there are details missing throughout, and the overarching hypothesis is not thoroughly tested through these studies. This is due to the pleiotropic nature of vitamin C actions, which are partially addressed by in vitro and conditional knockout studies. The authors note the impact of vitamin C on IOP as a confounding factor, but I feel a more important consideration is potential direct neuroprotective effects, bypassing the astrocytic pathway. This interpretation is supported by the author's own data showing induced neuronal expression of SPP1, but is not given much consideration in the present manuscript. Additional specific points are described below:

We agree that not all neuroprotective effects of vitamin C are mediated by the astrocytic release of SPP1. In addition, there are SPP1-positive retinal alpha ganglion cells, too. We have added paragraphs to the Discussion that discuss the effects of vitamin C on the IOP and on retinal ganglion cells in more detail.

1. What is the rationale for focusing on NaAscorbate treatment in vitro vs Vitamin C? This should be spelled out.

We have used vitamin C and its sodium, sodium ascorbate (NaAscorbate), in vitro in cultured astrocytes. Vitamin C is a weak acid and may cause change in gene transcription due to the effects of pH in cultured astrocytes. So we used NaAscorbate with the pH of the cell culture medium which was 7.40-7.47 for concentrations between 31.25-1000 μ M, excluding the effects of pH on gene transcription in astrocytes (Figure 1M). This is now spelled out in the Results.

2. Figure 1. What is the "n" for C, G, K panels? Also, the figure lists n=number of cells for the analyses, but please clarify if this was from multiple independent cultures/wells? Also, this is a minor point, but the presentation of panel M exaggerates the small pH differences at first glance,

I would suggest a revised presentation that shows the full scale (which will demonstrate the opposite impression)

We have added more information to the figure legend: The dose–response curves in Figure 1 C, 1G and 1K were made based on the SPP1 expression in Figure 1B, 1F and 1J separately. So the number in Figure 1 C, 1G and 1K is the same as that in Figure 1B, 1F and 1J. We have labeled the "n" in the figure legend for Figure 1 C, 1G and 1K. The cells used for analyses were from multiple independent wells. We are now using the full scale to show the pH of medium with different concentrations of NaAscorbate in Figure 1M.

3. Figure 2. Where do these data come from? There is no description in the methods and it seems unlikely that 48 genes were assessed by standard qPCR (if they were, then the primer sets should be attached as a supplemental file). What do the asterisks indicate in panel A? If this indicates significance, then it should be spelled out.

The 48 genes were tested in qPCR. We have added this information in the Results section and figure legends. The primer sequences are given in the Supplemental Table. This table was apparently missing from our original submission. We apologize for the oversight. The asterisks indicate in panel A indicate significance with $P < 0.05$ between vehicle and NaAscorbate in cultured C57BL/6 astrocytes, and between NaAscorbate-treated C57BL/6 and NaAscorbate-treated Spp1 KO astrocytes. This information has been added to the figure legend.

4. Figure 2. The text describing these data is a little too conclusive regarding A2 vs A1 astrocytic genes, as all seem impacted by both NaAscorbate treatment, and especially Spp1 KO. I suggest tempering this language to better describe the results.

We agree. Vitamin C does not switch astrocytes into a neuroprotective state. We see a stronger up-regulation of A2 genes than of A1 genes with vitamin C, suggesting a more neuroprotective phenotype. In the Spp1 KO astrocytes, A2 markers are down-regulated compared to the wild-type, and vitamin C does not reverse that, which suggests that the vitamin C effect on astrocyte gene expression is at least in part dependent on the presence of SPP1. We have modified the description to reflect this.

5. Figure 3. Unless I'm misunderstanding, it seems SPP1 is only staining in alpha ganglion cells in the retina in situ, and not astrocytes. What about the optic nerve? Doesn't this suggest that there might be direct neuroprotective effects for this treatment? This should be addressed. Also, how many animals do these data represent? (only #'s of analyzed cells are listed).

In the normal retina and optic nerve, alpha retinal ganglion cells are the only cells that express detectable levels of SPP1. After injury, SPP1 becomes detectable in astrocytes and microglia in the optic nerve. Alpha cells, too, up-regulate SPP1 after injury. This raises the possibility that alpha cells may also release SPP1, maybe as part of a protective response. Though this is certainly an attractive hypothesis, at the moment we do not have direct data to support it. We have added a paragraph to the Discussion section to address this effect. The analyzed alpha ganglion cells were from 3-4 mice. The numbers of mice have been added to the figure legend.

6. Figure 4. The colocalization of SPP1 staining in astrocytes of the ONH in panel A is not very convincing, although the sorting data do add alternative evidence. Once again, how were the 48-gene panel data generated?

The astrocytes in the ONH stain brightly for GFAP, but as it is a cytoskeletal protein, GFAP staining is not cell-filling and leaves some smaller processes unlabeled. In the ONH the situation is a bit simplified by the absence of oligodendrocytes and the scarcity of microglia. However, we would not make the claim that astrocytes express or up-regulate SPP1 on the basis of immunohistochemistry alone. Therefore, we also sorted astrocytes from GFAP-tdTomato mice by FACS to confirm Spp1 expression in astrocytes using qPCR (Figures 4C-E). We used qPCR for the experiments in Figure 4F, too. This information was added to the figure legend.

7. Figure 5. In the graphs B-E, it seems that the protective effect of vitamin C are independent of SPP1. The authors should comment on this observation and compare it to the optic nerve crush model, which does seem to be SPP1 dependent.

We agree that the IOP-lowering effect is not dependent on SPP1. At present, it is not known exactly how vitamin C lowers the IOP, but experimental evidence from the literature shows that trabecular meshwork cells respond to vitamin C, which may lead to an improved outflow of aqueous humor. We added a paragraph about this to the Discussion section.

8. In the discussion (note the missing heading), the authors focus on the potential for SPP1 to be induced in A2 reactive astrocytes. They also describe the potentially confounding effects of IOP lowering in the microbead model. However, it seems to me another clear possibility is that vitamin C induces direct neuroprotective signaling through RGCs themselves. Their own data in Figure 3 showing induced RGC expression of SPP1 supports this idea. There is also a fairly extensive literature linking vitamin C to a variety of neuroprotective effects. This point should be addressed - Otherwise, the conclusions about astrocytic functions should perhaps be tempered accordingly.

A heading to the Discussion was added. We certainly agree that not all neuroprotective effects of vitamin C are dependent on SPP1. We also hypothesize that vitamin C up-regulates SPP1 expression in alpha ganglion cells, and that this may be protective for the alpha cells themselves and possibly for neighboring neurons as well. We are now addressing these points directly in the Discussion.

Reviewer #2 (Comments to the Authors (Required)):

This manuscript describes the findings that vitamin C stimulates the astrocytic expression of the SPP1 (secreted phosphoprotein 1, osteopontin), thereby inducing a neuroprotective phenotype in astrocytes in vitro and in vivo. The authors used mouse models (wild type C57BL6) and Spp1 knock-out (B6.129S(Cg)-Spp1tm1Blh/J, 004936) as well as primary astrocyte cultures in their study. The manuscript is well-written and thoughtful, and the research is methodical, thorough,

and sound. With minor changes, publication of this manuscript would be valuable to the vision research community. Minor changes include:

-include all data points in dot format

We have included data points for most of the graphs. Due to the very high number of data points in the SPP intensity graphs, we have kept these graphs without data points.

-include representative traces of ERG data

Representative ERG traces were added as Supplementary Figure S2A.

-when introducing vitamin C, include (ascorbic acid) as vitamin A (retinoic acid) (page 5,7)

This sentence was indeed confusing and has been re-worded in the revised version.

-expand upon discussion on the results varied between NaAscorbate and VitC

The maximum effect on the increase of SPP1 expression was similar between vitamin C and sodium ascorbate, but the dose for the maximum effect was lower in sodium ascorbate (62.5 μ M) than that in vitamin C (125 μ M). We have added the discussion about the variance in the results.

Reviewer #3 (Comments to the Authors (Required)):

Referee Cross-Commenting:

Reviewers 1 and 2 raise some very valid points. I want to highlight the following points from reviewer # 1.

The Ns in some of the plots are missing and should be stated, e.g. in Fig 1 C, G, K. The number of animals/eyes should be stated for plots from *in vivo* experiments where only the number of cells are stated (e.g. in Fig 3).

We have included the N number in Figure 1C, 1G and 1K legends, and added mouse numbers for the *in vivo* experiments in the Figure 3 legends.

The supplemental data was missing from the merged manuscript pdf due to no fault of the authors I think. It does contain the primer sequences for the 48 genes that Reviewer #1 was looking for.

We have included a Supplementary Table showing the primer sequences.

1. A short summary of the paper, including description of the advance offered to the field.

In this manuscript, Li and Jakobs investigated the role of oral supplementation of vitamin C on preventing neurodegeneration and preserving visual function in mouse models of glaucoma. Using well designed, careful experiments they honed in on a particular mechanism which could be behind

the beneficial effects of vitamin C supplementation. Building on their previous work on the cytokine Spp1 (Li and Jakobs 2022), in this manuscript they show that vitamin C exerts a beneficial effect by inducing a neuroprotective class of reactive astrocytes both in well designed in vitro experiments as well as in in vivo models of glaucoma in mice. I think the paper is carefully written and the experiments in general are well designed and executed. The authors took care to utilize age, sex and strain matched controls in their in vivo experiments. I also really commend the authors for making the figures with great care. The data was very well presented. The discussion section of the paper is particularly well written and addresses important caveats including some that I thought of while reading the paper. The discussion on Vitamin C supplementation in humans vs. mice is really useful. I do think that the authors somewhat overstate the results for visual function in the abstract but overall its solid work. I think this paper certainly deserves to be published in Life Science Alliance, with the revisions to the text I requested in 3.

2. For each main point of the paper, please indicate if the data are strongly supportive. If not, explicitly state the additional experiments essential to support the claims made and the timeframe that these would require. The paper can be broadly divided into two subsections. The authors start out with crucial in vitro experiments that lay the foundation for their subsequent in vivo experiments in mice.

In the in vitro section, building on their prior work, the authors demonstrate that vitamin C as well as ascorbate can induce Spp1 expression in cultured astrocytes via the transcription factor E2f1 in a dose dependent manner (figure 1). The Spp1 induction in astrocytes is central to the authors' hypothesis on how Vitamin C exerts its protective effects, them having recently published a paper on this topic (Li and Jakobs 2022). The representative images in Figure 1 are convincing, the data is presented in a way that's easy to follow. In Figure 2, the authors utilize the findings from a landmark paper by Ben Barres' group in 2017 (Liddelow et al. 2017) showing that neuroinflammation and ischemia can induce two distinct category of reactive astrocytes, a neurotoxic class termed A1 and a neuroprotective class A2 and where class is characterized by their characteristic up/down regulation of key gene markers among other differences. In this manuscript, Li and Jakobs demonstrate that in vitro vitamin C supplementation induces markers in astrocytes that classify them closer to the A2 group of neuroprotective astrocytes in the Barres Lab paper.

In the next part of the paper, the authors investigate the protective role of Vitamin C in in vivo models of glaucoma in mice. The authors show that Vitamin C supplementation induces Spp1 expression in the retina, oddly enough in a subclass of retinal ganglion cells (RGCs): alpha. The fluorescence images look reasonably convincing. In Figure 4, the authors show that vitamin C oral supplementation induces Spp1 in RGCs and optic nerve head (ONH) astrocytes in an elevated IOP model in mice. Then, using an elegant technique the authors isolate the Spp1+ astrocytes from WT and Spp1 KO mice subjected to elevated intraocular pressure (IOP), and show that astrocytes in vitamin C treated mice but not vehicle treated mice exhibit elevated Spp1 as well as the transcription factor E2f1 in keeping with their results from in vitro experiments. Further, the astrocytes from Vitamin C treated IOP mice exhibit a gene expression profile that's putatively conducive to neuroprotection. This is also blocked in Spp1 KO mice showing that this effect of vitamin C in vivo in the glaucomatous mice is Spp1 dependent. This figure is really well presented and the experiments are very carefully designed and clever. In the next figure the authors seek to

demonstrate the beneficial effects of Vitamin C oral supplementation on the elevated IOP model in WT and SPP1 KO mice. Although Vitamin C appears to be neuroprotective (figure 5B) and appears to preserve visual function (figure 5C, 5D) in the IOP animal model in WT mice, interestingly the SPP1 KO mice subjected to IOP and treated with vitamin C also appear to benefit from a neuroprotective effect (figure 5B) and are able to preserve visual function (5C, 5D). The authors discovered that Vitamin C also causes a downregulation of IOP (figure 5E) and this thus presents a confounding factor. They navigate this issue by utilizing another model of RGC loss that is IOP independent, optic nerve crush (ONC). In figure 5G, even though the WT mice treated with vitamin C have higher RGC counts than vehicle, the Spp1 KO mice do not exhibit any difference between Vitamin C and vehicle treated animals. The authors state that vitamin C also reduced the IOP which could be responsible for the beneficial effects rather than Spp1 induction. It might be helpful to highlight here or in the discussion if this could potentially explain why the Spp1 KO mice in Figure 5B, 5C, 5D show a similar effect of Vitamin C treatment as the WT mice.

We agree that the IOP-lowering effect of vitamin C is not mediated by SPP1 as it occurs in the KO animals as well. Vitamin C is known to act on trabecular meshwork cells in the anterior chamber of the eye and improve outflow facility. This effect may account for the IOP lowering. We have added a paragraph discussing this possibility to the Discussion.

3. Lastly, indicate any additional issues you feel should be addressed (text changes, data presentation, statistics etc.). Important:

a. The sentence in the abstract (lines 5 through 7) "Here we show that vitamin C promotes a neuroprotective phenotype in reactive astrocytes by increasing the release of neurotrophic factors, phagocytosis, and mitochondrial ATP production." overstates what the authors have done in this manuscript. I appreciate that the authors did functional assays for mitochondrial ATP production and phagocytosis in their previous paper on Spp1 (Li and Jakobs 2022). But this sentence misleadingly gives the impression that similar work was performed in the current manuscript when the only experiments presented in this work are gene expression assays for markers of mitochondrial function and phagocytosis. I do not think that the authors need to do the functional experiments. However, the sentence in the abstract should be revised to accurately reflect the work that was done in this specific paper.

We revised the description about the effects of vitamin C on neurotrophic factors, phagocytosis, and mitochondrial ATP production in the Abstract section.

b. I commend the authors for discussing some of the caveats of the effect of Vitamin C on IOP. However, I do think that based on the constraints of the experiments in figure 5, the authors cannot conclude that Spp1+ astrocytes are causing preservation of visual function. This is especially because the Spp1 KO mice in Figure 5C and 5D also exhibited improved visual function when treated with Vitamin C. If the preservation of visual function by Vitamin C is Spp1+ astrocyte dependent then the data from Spp1 KO mice in figures 5C and 5D contradict this claim. The only beneficial effect they can conclude for certain is the neuroprotective effect based on the Brn3a+ RGC count in figure 5G where the KO mice do not show any effect due to Vitamin C supplementation. As visual function is not assessed in the ONC model after vitamin C supplementation, the claim about visual function should be proportionate to the results presented.

Thus, the appropriate parts of the manuscript should be revised to reflect this. i.e., in the abstract (lines 9 through 10): "SPP1+ astrocytes in turn promote RGC survival and improve visual function in a mouse model of glaucoma.". They can revise it to say that "Vitamin C promotes RGC survival and visual function" or "SPP1+ astrocytes in turn promote RGC survival."

We revised the Abstract and Discussion accordingly.

c. It would be helpful if the authors can include in the methods/figure legends more details about how the fluorescence intensities were quantified and plotted, especially for the *in vivo* experiments like in figure 3, whether the Spp1 intensities per cell for all SMI32+ cells in the whole retina were quantified or some parts (which parts) etc. Assuming that the intensity plotted is per cell and the error bars are over cells, it would be important to know these details in order to better assess how robust the effect of Vitamin C or ascorbate supplementation is on a per animal basis. Moreover, it would be great if they could clarify how they handled Spp1 negative cells in this quantification. Were Spp1 negative cells included in the calculation? So lets say, hypothetically speaking a cell was negative for Spp1 in the vitamin C treatment group (even though it is SMI32+) would that have been included in the means calculation for the group? In figure 3E, the legend says n = 29-103 cells/group. It would be very helpful to state how the authors chose how many, which cells to count, in the whole mount retina for plots like this. Obviously, based on the numbers it does not look like the whole retina was quantified(?).

We have included more details about the quantification of SPP1 fluorescence intensity in the Methods section. SPP1 protein expression in retina *in vivo* was quantified by the fluorescence intensities of SPP1 immunostaining in SPP1 positive retinal ganglion cells. SPP1 negative RGCs were not quantified. SPP1-positive RGCs in the acquired images from parts of the retina were used for quantification; and fluorescence intensities of SPP1 immunostaining were analyzed in the plotted region per cell with ImageJ software. Mean fluorescence of SPP1 immunostaining in all quantified RGCs was recorded and used for comparison.

Minor comments:

a. Do both OFF and ON alpha RGCs express Spp1? What, if any is the role of Spp1 in RGCs?

SPP1 is expressed in ON and OFF alpha cells. It is not known what the physiological role of SPP1 in these cells may be. In normal retinas, SPP1 staining is concentrated in the perinuclear region, possibly in the ER or Golgi. SPP1 does not seem to be associated with typically neuronal structures such as dendrites, synapses, or axons. This is now mentioned in the Discussion.

b. An RNAseq approach would have been better for figure 4f, especially since it may have helped generate more insight about the Vitamin C induced, neuroprotection mechanism downstream of Spp1 and potentially also generated putative small molecule targets. RNAseq on retinal cells after Vitamin C treatment in Spp1 KO vs WT animals might also be useful for future studies.

Yes, it would indeed be very useful to perform these RNA-seq studies in astrocytes and retinal ganglion cells treated with vitamin C. We are planning to do these experiments in the future.

April 21, 2023

RE: Life Science Alliance Manuscript #LSA-2023-01976-TR

Dr. Tatjana C Jakobs
Massachusetts Eye and Ear Infirmary
Ophthalmology
20 Stanford Str
Boston, MA 2114

Dear Dr. Jakobs,

Thank you for submitting your revised manuscript entitled "Vitamin C protects retinal ganglion cells via SPP1 in glaucoma and after optic nerve damage". We would be happy to publish your paper in Life Science Alliance pending final revisions necessary to meet our formatting guidelines.

- please address Reviewer 3's remaining comments
- please upload your table files as editable doc or excel files
- please add ORCID ID for corresponding author-you should have received instructions on how to do so
- please add the Twitter handle of your host institute/organization as well as your own or/and one of the authors in our system
- please use the [10 author names, et al.] format in your references (i.e. limit the author names to the first 10)
- please remove the figure callout for Figure S2A-since it's the only panel in the figure, we don't need it designated with a letter; please adjust the figure callouts accordingly and remove the panel in the figure itself

Figure Check:

- you may consider uploading Figure 6 as a Graphical Abstract instead of as a figure, but this is up to you

A. FINAL FILES:

B. MANUSCRIPT ORGANIZATION AND FORMATTING:

Sincerely,

Reviewer #1 (Comments to the Authors (Required)):

The authors have made a good effort to address the reviewer points, and in particular the additional experimental details and discussion have provided clarification. No further revisions are requested.

Reviewer #3 (Comments to the Authors (Required)):

The authors have addressed most of my concerns about the paper that I raised in my previous comments.

However, in the revised manuscript I noticed an irregularity that should be urgently addressed before I can recommend publication of this to go ahead.

In the supplementary figure that the authors added in the revised manuscript, i.e. figure S2 <2520_1_unknown_upload_647983_rs4hlm.pdf or.tiff>:

(1) The pERG representative trace labeled "IOP + vehicle" appears to be a carbon copy (!) of the trace labeled "Spp1 +/- IOP + Vitamin C".

(2) The pERG representative trace labeled "IOP + Vitamin C" appears to be a carbon copy (!) of the representative trace labeled to be that of Spp1 +/- mice in Li and Jakobs 2022, figure S2 C, extended pdf page 30 [https://www.cell.com/cell-reports/pdfExtended/S2211-1247\(22\)01776-4](https://www.cell.com/cell-reports/pdfExtended/S2211-1247(22)01776-4). Either, the trace in the authors' earlier paper is not what it says in that paper or the wrong trace was submitted with this manuscript.

I checked the traces in question carefully and the waveforms are exactly the same for the ones I highlighted above. Unless they are exactly the same recording, its virtually impossible for them to be that similar.

I hope the authors will check the raw data underlying this figure and submit the correct traces in an updated figure.

We thank the reviewers for their comments. Our responses are underlined.

-please address Reviewer 3's remaining comments

Please see below.

-please upload your table files as editable doc or excel files

-please add ORCID ID for corresponding author-you should have received instructions on how to do so

Done as requested.

-please add the Twitter handle of your host institute/organization as well as your own or/and one of the authors in our system

The Twitter handle of our institution is @MassEyeAndEar. Neither of the authors is on Twitter.

-please use the [10 author names, et al.] format in your references (i.e. limit the author names to the first 10)

Done as requested.

-please remove the figure callout for Figure S2A-since it's the only panel in the figure, we don't need it designated with a letter; please adjust the figure callouts accordingly and remove the panel in the figure itself

Done as requested.

Figure Check:

-you may consider uploading Figure 6 as a Graphical Abstract instead of as a figure, but this is up to you

We would prefer to leave it as Figure 6.

LSA now encourages authors to provide a 30-60 second video where the study is briefly explained. We will use these videos on social media to promote the published paper and the presenting author (for examples, see

<https://twitter.com/LSAjournal/timelines/1437405065917124608>). Corresponding or first-authors are welcome to submit the video. Please submit only one video per manuscript. The video can be emailed to contact@life-science-alliance.org

A. FINAL FILES:

B. MANUSCRIPT ORGANIZATION AND FORMATTING:

Sincerely,

Reviewer #1 (Comments to the Authors (Required)):

The authors have made a good effort to address the reviewer points, and in particular the additional experimental details and discussion have provided clarification. No further revisions are requested.

Thank you very much.

Reviewer #3 (Comments to the Authors (Required)):

The authors have addressed most of my concerns about the paper that I raised in my previous comments.

However, in the revised manuscript I noticed an irregularity that should be urgently addressed before I can recommend publication of this to go ahead.

In the supplementary figure that the authors added in the revised manuscript, i.e. figure S2 <2520_1_unknown_upload_647983_rs4hlm.pdf or.tiff>:

(1) The pERG representative trace labeled "IOP + vehicle" appears to be a carbon copy (!) of the trace labeled "Spp1 -/- IOP + Vitamin C".

We have corrected this mistake. Please see comment to (2) for a description of how that happened.

(2) The pERG representative trace labeled "IOP + Vitamin C" appears to be a carbon copy (!) of the representative trace labeled to be that of Spp1 -/- mice in Li and Jakobs 2022, figure S2 C, extended pdf page 30 [https://www.cell.com/cell-reports/pdfExtended/S2211-1247\(22\)01776-4](https://www.cell.com/cell-reports/pdfExtended/S2211-1247(22)01776-4). Either, the trace in the authors' earlier paper is not what it says in that paper or the wrong trace was submitted with this manuscript.

I checked the traces in question carefully and the waveforms are exactly the same for the ones I highlighted above. Unless they are exactly the same recording, its virtually impossible for them to be that similar.

I hope the authors will check the raw data underlying this figure and submit the correct traces in an updated figure.

We thank the reviewer for these comments. The traces were indeed identical. The reason is that we made a template for preparing the ERG figure in the previous Cell Reports manuscript. This was done because it saves time for the sizing and positioning of panels. We re-used the template for the present manuscript, and then forgot to delete the layers with the old traces when we put together Figure S2. Neither of us noticed that in time. We have now supplied the correct traces. Furthermore, we have uploaded all original traces to the Harvard Dataverse as "Vitamin C and SPP1 in glaucoma" (<https://doi.org/10.7910/DVN/WTEOGO>). For all cases, trace 1 (of 8 traces per experimental group) was used to prepare Figure S2. Going forward, we will completely end the use of figure templates, as it is obviously too error-prone. Again, thank you very much for catching this mistake.

May 1, 2023

RE: Life Science Alliance Manuscript #LSA-2023-01976-TRR

Dr. Tatjana C Jakobs
Massachusetts Eye and Ear Infirmary
Ophthalmology
20 Stanford Str
Boston, MA 2114

Dear Dr. Jakobs,

Thank you for submitting your Research Article entitled "Vitamin C protects retinal ganglion cells via SPP1 in glaucoma and after optic nerve damage". It is a pleasure to let you know that your manuscript is now accepted for publication in Life Science Alliance. Congratulations on this interesting work.

DISTRIBUTION OF MATERIALS:

Again, congratulations on a very nice paper. I hope you found the review process to be constructive and are pleased with how the manuscript was handled editorially. We look forward to future exciting submissions from your lab.

Sincerely,
